# Lyme Carditis: From Pathophysiology to Clinical Management

**DOI:** 10.3390/pathogens11050582

**Published:** 2022-05-15

**Authors:** Cinzia Radesich, Eva Del Mestre, Kristen Medo, Giancarlo Vitrella, Paolo Manca, Mario Chiatto, Matteo Castrichini, Gianfranco Sinagra

**Affiliations:** 1Center for Diagnosis and Treatment of Cardiomyopathies, Cardiovascular Department, Azienda Sanitaria Universitaria Giuliano Isontina (ASUGI) and University of Trieste, 34127 Trieste, Italy; c.radesich@gmail.com (C.R.); eva.delmestre@gmail.com (E.D.M.); giancarlo.vitrella@gmail.com (G.V.); paolo.manca91@yahoo.it (P.M.); gianfranco.sinagra@asugi.sanita.fvg.it (G.S.); 2Division of Cardiology, Anschutz Medical Campus, University of Colorado, Aurora, CO 80045, USA; kristen.medo@cuanschutz.edu; 3Interventional Cardiology Department, Ospedale Civile dell’Annunziata, 87100 Cosenza, Italy; compama@libero.it

**Keywords:** Lyme carditis, Lyme disease, *Borrelia burgdorferi*, atrioventricular block, temporary pacing, doxycycline

## Abstract

Cardiac involvement is a rare but relevant manifestation of Lyme disease that frequently presents as atrioventricular block (AVB). Immune-mediated injury has been implicated in the pathogenesis of Lyme carditis due to possible cross-reaction between *Borrelia burgdorferi* antigens and cardiac epitopes. The degree of the AVB can fluctuate rapidly, with two-thirds of patients progressing to complete AVB. Thus, continuous heart rhythm monitoring is essential, and a temporary pacemaker may be necessary. Routinely permanent pacemaker implantation, however, is contraindicated because of the frequent transient nature of the condition. Antibiotic therapy should be initiated as soon as the clinical suspicion of Lyme carditis arises to reduce the duration of the disease and minimize the risk of complications. Diagnosis is challenging and is based on geographical epidemiology, clinical history, signs and symptoms, serological testing, ECG and echocardiographic findings, and exclusion of other pathologies. This paper aims to explain the pathophysiological basis of Lyme carditis, describe its clinical features, and delineate the treatment principles.

## 1. Introduction

Lyme disease is an infectious disease caused by the spirochete *Borrelia burgdorferi* and is transmitted through the bite of infected hard-bodied ticks in the genus Ixodes. During the early disseminated phase, the spirochete spreads into different organs and may include cardiac involvement, which presents predominantly as high degree atrioventricular block (AVB). While Lyme carditis is a rare event, the incidence of Lyme disease is increasing [1] so more cases may be seen in the future. More importantly, prompt recognition of this illness is crucial due to its reversible nature and to avoid an unneeded permanent pacemaker implantation.

## 2. Lyme Disease: Pathogen, Transmission, and Clinical Manifestations

The *B. burgdorferi* sensu latu (s.l.) complex currently comprises 23 species of Borrelia, 9 of which (*B. burgdorferi* sensu stricto, *B. afzelii*, *B. bavarensis*, *B. bissettii*, *B. garinii*, *B. lusitaniae*, *B. mayonii*, *B. spielmani*, *B. valaisiana*) can cause infection in humans [2]. These Borrelias are organotropic and only *B. mayonii*, recently isolated in the USA, is spirochetemic [3]. *B. burgdorferi* is the most prevalent species in the United States and is mostly arthritogenic, whereas *B. afzelii* and *B. garinii* are the dominant species in Europe and Asia, and can cause skin lesions and neurological syndromes, respectively [4].

*B. burgdorferi* s.l. is transmitted by *Ixodes* spp. ticks (in particular, by *I. scapularis* and *I. pacificus* in eastern and western North America, respectively, by *I. ricinus* in Europe and by *I. persulcatus* in Asia) [2]. The tick lifecycle consists of four stages: egg, larva, nymph, and adult. Larvae of *I. ricinus* feed on infected reservoir animals and then molt into nymphs. Nymphs are primarily responsible for the transmission of Lyme disease to humans, who are accidental hosts. Adult ticks mate on deer, which are essential to preserve the tick population, even though they are not considered a reservoir for the spirochetes [4]. The transovarian transmission of Borrelia Lyme Group is rare in the ticks [5].

The main reservoirs of *B. burgdorferi* are small rodents (*Peromyscus leucopus, Tamiasciurus hudsonicus, Sciurus criseus*) in the United States and small rodents (*Apodemus flavicollis, Myodes glareolus*) and birds (*Turdus merula*) in Europe [2]. Of note, migratory birds are responsible for dispersing ticks in new areas [4].

Spirochetes are transmitted to humans via the saliva of the ticks at the site of the bite, and infection is more likely if the tick remains attached to the skin for at least 48 h, as the bacteria must travel from the tick’s gut to its salivary glands [6,7]. After pathogen transmission to humans, Lyme disease develops in three stages: early localized, early dissemination, and late dissemination. The first is the early localized phase, characterized by the appearance of erythema migrans within 3 to 32 days of transmission, usually accompanied by constitutional symptoms in the United States, but not in Europe. The spirochetes then spread to various organs through the circulatory and lymphatic systems, potentially causing dermatologic, joint, cardiac, and neurological symptoms. This is known as the early dissemination phase and occurs days to weeks after the tick bite. The last stage, the late dissemination phase, occurs after approximately 2–3 years after pathogen transmission (or even later) and includes chronic arthritis of one or few large joints (especially in the United States) and *acrodermatitis chronica atrophicans* (especially in Europe, and primarily by *B. afzelii*) [6,7].

## 3. Epidemiology of Lyme Disease and Lyme Carditis

Lyme disease is spread throughout the Northern Hemisphere, and both its incidence and its geographical range are increasing. These epidemiological trends can be explained, among other reasons, by climate-mediated improvement of tick survival and by environmental changes, such as the reforestation of former agricultural lands, which leads to a greater prevalence of vertebrate reservoirs and host deer [1]. Current data shows Lyme borreliosis represents the most prevalent vector-borne illness in the United States and in Europe with an estimated number of cases of 476,000 and 85,000 per year, respectively [7,8]. Most cases of Lyme disease happen between June and July due to the feeding habits of nymphal ticks [2].

Lyme carditis is a rare event, with an estimated incidence in untreated adults of 0.3–4% in Europe and 4–10% in the United States [9]. However, more recent data suggest the incidence in the United States may be as low as 1% [10,11], likely due to better recognition and prompt treatment of Lyme disease [12]. It has been suggested that the difference in the observed incidence between Europe and the United States could be explained by the presence of less cardiotropic genospecies of *B. burgdorferi* s.l. in Europe, such as *B. garinii* and *B. afzelii*, compared to *B. burgdorferi*, the most predominant species in the United States [13].

Lyme carditis occurs more frequently from June to December due to vector, reservoir, and host behavior, and symptoms can appear from a few days to seven months after the tick bite or the appearance of erythema migrans [11]. However, just 40% of patients recall the presence of skin lesions [14].

Cardiac involvement may be isolated, but more commonly is accompanied by cutaneous (erythema migrans), joint (arthritis), or neurologic (neuroborreliosis) manifestations [9,15].

Despite the incidence of Lyme disease being almost equal in the two sexes (with just a slight male preponderance), Lyme carditis is predominant in men, with a sex ratio of approximately 3:1 [16,17]. Additionally, people aged 20–40 years are more likely to be affected [16].

## 4. Pathogenesis and Histology

The spirochetes must survive both in ticks and in mammalian hosts. Ticks do not thermoregulate and feed once or twice per year, so the spirochetes must adapt to a wide range of temperatures and survive with a paucity of nutrients. This is particularly remarkable, as *B. burgdorferi*’s genome does not encode for essential products, such as amino acids, fatty acids, enzyme cofactors, and nucleotides, resulting in an organism completely dependent upon the host [4].

Once injected into the human skin, spirochetes replicate locally and then disseminate into the organism, persisting in different tissues. Mammals offer a stable temperature and abundant nutrients; however, they are capable of a sophisticated immune response.

A key mechanism for *B. burgdorferi* to resist both innate and adaptive immunity is regulating the expression of its surface proteins. For example, the bacterium can resist complement-mediated killing through the varied expression of surface proteins (Outer surface protein E-related proteins and Complement regulator-acquiring surface proteins), that bind factor H and complement factor H-like protein. These factors bind and inhibit C3b, thus inactivating the complement cascade [4]. Additionally, to evade the adaptive immune system, *B. burgdorferi* decreases the production of Outer surface protein C (OspC) and upregulates the expression of the Variable Lipoprotein Surface-Exposed protein (VlsE). This mechanism is essential, as IgM antibodies against OspC are produced very early by the organism and can kill the bacteria. VlsE likely has the same physiological function as OspC but can undergo antigenic variation, thus eluding the host immune response [4].

### 4.1. Pathogenesis of Cardiac Damage and Histological Findings

To colonize different tissues, *B. burgdorferi* must modulate the expression of its surface proteins to gain the ability to bind to host tissue cells, extracellular matrix, and vasculature. In particular, some borrelial proteins have been identified as critical for cardiac tropism, such as P66 and decorin-binding protein.

P66 is an important borrelial protein that binds to integrin receptors and seems to promote the dissemination of the bacteria into the bloodstream as well as cardiac tropism. Indeed, Caine et al. have shown in an in vivo murine model that a strain of *B. burgdorferi* deficient in the expression of the integrin-binding adhesin P66 is less able to colonize the heart [18]. Interaction with decorin, a ubiquitous proteoglycan of the extracellular matrix, has also proven essential to *B. burgdorferi*’s heart colonization [19] Furthermore, allelic variation of genes encoding for the bacteria’s surface protein, such as decorin-binding adhesin A, may explain the distinct tissue tropism of different species and strains of *Borrelia*, possibly causing different clinical manifestations [20]. Of note, spirochetal interaction with host decorin may also be protective against both innate and adaptive immune responses [21,22].

Once the myocardium is colonized, an exaggerated immune response is induced, causing cardiac injury. To corroborate this assumption, the lymphocytic infiltration in Lyme carditis, both in humans and in animals, is usually more predominant compared to the presence of the spirochetes [23,24,25,26]. Additionally, murine models have demonstrated an autoimmunity response can be triggered against the heart by IgM antibodies directed against *B. burgdorferi* antigens. This phenomenon may occur due to a cross-reaction between antigens of the Spirochetes and endogenous cardiac proteins [27].

### 4.2. Histological Findings

In animal models, inflammation observed in Lyme carditis is typically transmural, presenting as a band-like infiltration of macrophages and lymphocytes, with or without the presence of the pathogen [25]. In contrast, in Lyme arthritis, a predominant neutrophil infiltration affects the joints [28]. These histologic findings are more frequently detected in the connective tissue at the base of the heart, interventricular septum, and perivascular areas [26,29,30]. This manifestation may justify the primary clinical presentation observed in Lyme carditis: heart rhythm abnormalities, especially AVB [24]. Furthermore, *B. burgdorferi* relies only on glycolysis for energy production and myocytes of the atrioventricular node contain numerous glycogen granules, unlike contracting cardiomyocytes, which depend on fatty acid oxidation [31].

Autoptic studies of people who died secondary to Lyme pancarditis have shown diffuse and severe interstitial infiltrates affecting all areas and layers of the heart, with a predilection for perivascular regions. Spirochetes were observed along collagen fibers and decorin protein. Interestingly, other tissues expressing decorin were not colonized. To explain this phenomenon, the authors suggest cardiac-specific modifications of glycosaminoglycan groups of the protein may promote bacterial adhesion in the heart. Inflammatory infiltrates consisted mainly of lymphocytes, histiocytes, and plasma cells. Granulomas, vasculitis, or significant eosinophilic infiltrates were not observed. Coronary arteries were unscathed. In contrast, infiltrates were noted in the conduction system, and, in one case, the atrioventricular node showed prominent necrotizing inflammation [32,33].

## 5. Clinical Manifestations

### 5.1. Conduction Abnormalities

Clinical manifestations of Lyme carditis are heterogeneous and non-specific, ranging from asymptomatic forms to arrhythmic syncope. Patients can experience symptoms such as dyspnea, lightheadedness, palpitations, or chest pain [9]. Signs and symptoms of cardiac involvement are usually preceded in the previous weeks by skin lesions or arthralgias [34]. Rarely, Lyme carditis may be lethal [32].

In up to 90% of cases, Lyme carditis presents as AVB [24]. AVB determined by Lyme disease is typically intermittent and of fluctuating gravity, shifting from first-degree to third-degree AVB (and vice versa) in days, hours, or even minutes. First-degree AVB reflects prolonged atrioventricular conduction and consists of a PR interval greater than 200 milliseconds (msec). Second-degree AVB is caused by the dysfunction of the AV node or the His–Purkinje system and its manifestation is represented by a non-conducted P wave. Second-degree AVB can be divided into two types, Mobitz type 1 or Luciani–Wenckebach and Mobitz type II. Mobitz type 1 or Luciani–Wenckebach is characterized by a progressive lengthening of the PR interval until a P wave is not conducted. Mobitz type II consists of occasional non-conducted P waves, without the lengthening of the PR interval. Third-degree AVB is caused by a complete failure of the AV node or the His–Purkinje system; its manifestation on the ECG is the dissociation of atria and ventricles and an escape junctional or ventricular rhythm.

An example of third-degree AVB is shown in Figure 1**.** Remarkably, complete AVB occurs in up to two-thirds of cases [35]. Thus, even seemingly reassuring clinical presentations should undergo a thorough electrocardiogram (ECG) and telemetry monitoring. Notably, patients presenting with a PR interval longer than 300 msec are at greater risk of evolution towards a high-degree AVB [36].

Differential diagnosis of AVB must include medications (especially antiarrhythmic and neuroactive drugs), ischemic etiologies (anterior and inferior myocardial infarction), degenerative diseases (e.g., Lev–Lenègre syndrome), cardiomyopathies (such as cardiomyopathy induced by lamin A/C mutations), other infections (such as endocarditis with perivalvular abscesses, Chagas disease), infiltrative processes (sarcoidosis, amyloidosis), and metabolic and neuromuscular disorders (myotonic dystrophy, Emery–Dreyfuss, and limb-girdle muscular dystrophies) [37]. A comprehensive list is shown in Table 1.

Lyme carditis conduction abnormalities are usually supra-Hisian and involve the atrioventricular node; however, other conduction abnormalities may occur, such as sinoatrial block, bundle branch block, and asystole due to transient failure of an escape rhythm [16]. The above-mentioned conduction abnormalities are transient, particularly once the infectious etiology of the condition is recognized and prompt antibiotic therapy is initiated [35].

Additional arrhythmias have been described in Lyme carditis, such as supraventricular tachycardia, atrial fibrillation, ventricular tachycardia, and ventricular fibrillation [24].

### 5.2. Endocarditis, Myocarditis, and Pericarditis

In a minority of cases, cardiac involvement in Lyme disease may present as endocarditis, myocarditis, pericarditis, or pancarditis [24].

Lyme endocarditis is anecdotal and can present identically to other forms of infective endocarditis. In this case, borreliosis infection should be suspected, especially in endemic areas, based on the clinical history and corroborated by etiological findings [38]. However, the diagnosis can be extremely difficult if endocarditis is the only manifestation of Lyme disease, since serology cannot distinguish between current and prior infection, and culture has a low sensitivity and many limitations. If surgery is required and there is suspicion of infectious etiology, a myocardial tissue sample should be obtained during the procedure to undergo histopathological, cultural, and PCR analysis [39].

Myopericarditis is typically self-limited, and patients are commonly asymptomatic. However, symptoms may include chest pain, non-specific ST-segment alterations or T wave abnormalities, and elevation of biomarkers of myocardial injury, hence mimicking an acute coronary syndrome [11,40]. Pericardial effusion can be observed and, occasionally, cardiac dysfunction may be present [7].

### 5.3. Dilated Cardiomyopathy

As for myocardial involvement, a potential correlation between *Borrelia burgdorferi* and dilated cardiomyopathy (DCM) is controversial. In central Europe, a possible association has been suggested and response to specific antibiotic therapy among patients with concurrent DCM and *B. burgdorferi* infection has been reported [41], while these findings have been inconsistent or absent in the United States and United Kingdom studies [42,43]. Thus, in the United States, routine testing for Lyme disease in patients with DCM of unknown cause is currently not recommended [14].

## 6. Diagnosis

The diagnostic pathway that leads to Lyme carditis should integrate different aspects, such as the epidemiology of the geographic area, clinical presentation (high suspicion should arise when patients under the age of 50 manifest atrioventricular conduction abnormalities), and exclusion of other pathologies and serological testing for Borrelia. Diagnosis is facilitated by the coexistence of dermatologic, joint, or neurologic symptoms, while it is particularly difficult in cases of isolated carditis. If Lyme carditis is hypothesized to be the cause of a high-degree AVB, the Suspicious Index in Lyme Carditis (SILC) score may aid in the diagnosis. This novel risk score is comprised of six variables represented by constitutional symptoms (2 points), history of outdoor activity or endemic area (1 point), male sex (1 point), history of a tick bite (3 points), age less than 50 years (1 point), presence of the pathognomonic erythema migrans (4 points). The six variables may be better remembered with the mnemonic formula “CO-STAR” (Constitutional symptoms, Outdoor activity/endemic area, Sex, Tick bite, Age, Rash) [44]. The score allocates patients into three categories: low (0–2), medium (3–6), and high risk (7–12). In medium- and high-risk cases, serological tests are obtained, and empiric antibiotic therapy is started [24], as shown in the summary scheme in Figure 2.

### 6.1. Serologic Testing

The primary immune response in Lyme disease is characterized by the appearance of IgM within one to two weeks and of IgG within two to six weeks following the onset of erythema migrans [45]. However, only 20–40% of patients with early localized Lyme disease are seropositive at the time of presentation [45,46] and serological testing is not required in patients presenting with a clinical diagnosis of erythema migrans (especially single, but also multiple erythemas) [14]. During the early disseminated phase, most patients present both IgM and IgG seropositivity against *B. burgdorferi*, and serological tests are helpful to support the diagnosis. Eventually, at the time of the late disseminated phase, IgM tests may be negative, while IgG response is usually present [47]. Lyme carditis occurs during the early disseminated phase, so patients are usually seropositive by the time of evaluation. Patients with a negative result are unlikely to have Lyme disease; however, if antibiotic therapy has not been initiated and the clinical suspicion persists, serologic testing may be repeated approximately three weeks later [45].

To prove seropositivity for Lyme disease, a two-tiered testing strategy is mandatory, using a traditional or a modified algorithm. The traditional method includes a first analysis using a sensitive enzyme immunoassay (EIA), such as a whole cell-based enzyme-linked immunosorbent assay (ELISA), or an immunofluorescence assay (IFA).

In cases of a positive first test, a confirmation test is performed using a Western Blot, which is more specific. According to the Centers for Disease Control and Prevention (CDC), the Western blot is positive if at least 2 of 3 bands are present on the IgM immunoblot within 30 days of symptom onset or 5–10 bands are present on the IgG immunoblot at any time. The diagnostic algorithm in Europe is not standardized due to antigen variability and differences in laboratory equipment [48,49,50].

Alternatively, modified algorithms include the use of two sequential EIAs with different targets. This new strategy is suggested in cases of early disseminated Lyme diseases (and therefore in Lyme carditis), as it offers better sensitivity and specificity [51].

Serological testing is readily available and an easy to perform tool, but it also presents some limitations, including the above-mentioned lack of sensitivity in the first phase and the interlaboratory variability. Additionally, serologic tests are not useful in cases of prior Lyme disease, since both IgG and IgM antibodies may persist for months or even years after *B. burgdorferi* infection [52].

As for the whole cell-based ELISA, since the antigen used is based on the lysates of the whole organism, false positives may occur due to a cross-reaction phenomenon, especially in patients with other infections (borrelial, spirochetal, bacterial, and viral diseases) or autoimmune illnesses [53,54]. It is estimated up to 5% of the population living in endemic areas is seropositive by ELISA [55]. False-positive results are also possible on Western Blot assays, especially IgM tests, so an isolated IgM Western blot after six to eight weeks of symptoms shall be considered a false positive [56].

For these reasons, serological testing alone is not sufficient to make nor exclude a diagnosis of Lyme disease, and it should not be used in patients with non-specific symptoms and a low pre-test probability. Most importantly, serologic results must always be correlated with the clinical presentation.

### 6.2. Other Laboratory Tests

Polymerase Chain Reaction (PCR) assay for Borrelia DNA detection in blood samples is difficult to perform due to transient spirochetemia. PCR analysis on endomyocardial biopsy specimens is recommended for exceptional cases, even if the absence of the microorganism does not rule out the diagnosis of Lyme carditis [7,57].

A blood culture is an alternative diagnostic test that allows a definitive diagnosis, but it is not routinely performed since it requires specific laboratory equipment and special expertise, restricting the technique only for research aims. Furthermore, Borrelia is a slow-growing bacterium with a long incubation time, which limits a culture’s contribution to clinical practice [4,48].

### 6.3. Cardiac Evaluation: Electrocardiogram, Echocardiogram, and Other Imaging Techniques

A 12-lead ECG and a 24-h Holter ECG should always be performed in cases of clinical suspicion of Lyme carditis. ECG alterations can be classified as non-specific, reflecting myocardial involvement, or specific, implying impairment of the conduction system [11].

An echocardiogram should also be performed to evaluate ventricular morphology, which may be normal or exhibit dilation of one or both ventricles [58]. Left ventricular function, as well as regional and global wall motion, are typically normal. Diffuse hypokinesia and depressed systolic function may be found in cases of myocarditis and/or pericarditis; however, most cases exhibit structural and functional abnormalities that are mild and transient [59].

Very few data are available about tissue characterization with Cardiac Magnetic Resonance (CMR) in Lyme myopericarditis. A thorough literature review provided only four case reports, one of which is a pediatric case [60]. Avitabile et al. reported signs of hyperemia and edema at T1 and T2 sequences, respectively [60]. In all the reported cases, late gadolinium enhancement (LGE) shows a subepicardial or mid-wall distribution (non-ischemic pattern), sparing the subendocardial layers [40,60,61]. Regions involved by LGE in different clinical scenarios were various and included ventricular septum, posterolateral, anterolateral, and inferior wall involvement, demonstrating how every region might be affected. In one case, CMR performed three months after the acute phase showed complete normalization of left ventricular size and function with no signs of delayed enhancement [62]. However, CMR in the acute phase was not available, thus precluding a comparison between the two moments. In the two cases where CMR was performed, both in the acute and recovery phase, LGE persisted [61] or it just partially resolved [40]. Due to few and divergent findings, more CMR data are needed to better characterize myocardial involvement in Lyme carditis, both during the acute phase and after the resolution of symptoms.

Updated literature about nuclear imaging in Lyme carditis is still missing; studies conducted during the 1980s and 1990s suggested a role of Gallium 67 scintigraphy and Indium 111-antimyosin scintigraphy in the diagnosis of Lyme myocarditis. Currently, nuclear imaging is not included in the diagnostic flow chart for Lyme carditis [63,64].

Endomyocardial biopsy is an invasive technique and should be indicated in selective cases, following the same guidelines for myocarditis protocols [65]. As correct antibiotic therapy can result in the complete resolution of the disease, it is reasonable to reserve this invasive test for non-responsive cases. A case report of the group of Reznick et al., in which a right endomyocardial biopsy was performed, reported a lymphoplasmacytic infiltration of the endocardium and perivascular myocardium, with varying degrees of myocyte injury [66].

## 7. Therapy

### 7.1. Antibiotic Prophylaxis

Chemoprophylaxis after a tick bite should be administered within 72 h only if the tick bite occurred in a highly endemic area or if the tick was attached for more than 36 h. The recommended antibiotic regimen is represented by a single dose of oral doxycycline (200 mg for adults and 4.4 mg/kg for children) [14].

### 7.2. Antibiotic Therapy

Lyme carditis can resolve spontaneously; however, appropriate antibiotic treatment decreases the duration of the disease and prevents complications [34]. Thus, antibiotic therapy must be started as soon as there is a clinical suspicion of Lyme carditis [24].

The choice of antibiotic regimen and method of administration depends on the severity of presentation and the risk of progression to a complete AVB. Therefore, treatment must be individualized based on patient presentation, considering age, hemodynamics, and peculiar features.

According to current American guidelines [14,67], adult patients with a mild Lyme carditis, defined as a first-degree AVB with a PR interval less than 300 msec, can be treated orally with doxycycline 100 mg, twice per day; amoxicillin 500 mg, three times per day; or cefuroxime 500 mg, twice per day. The same antibiotic regimens are used in children, adapting the dose to patient weight (for a more comprehensive scheme, see Table 2). It should be noted that doxycycline has better central nervous system penetration and is also effective against potential coinfecting pathogens, such as *Anaplasma phagocytophilum* [68].

Severe forms are characterized by the presence of symptoms (syncope, dyspnea, and chest pain), first-degree AVB with a PR interval greater than 300 msec, second- or third-degree AVB, or other arrhythmias. As expected, patients with higher degrees of AVB are usually more symptomatic [36]. In these cases, patients must be hospitalized to receive antibiotic treatment intravenously. First-line therapy is represented by intravenous ceftriaxone 2 g, once a day, while pediatric dosage is adapted to the weight (50–75 mg per kg, once a day). Once clinical improvement is obtained (e.g., resolution of high degree AVB or shortening of PR interval under 300 msec), antibiotic therapy can be switched to oral administration to complete the course, as the superiority of parenteral over oral therapy has not been demonstrated and intravenous therapy is associated with a greater potential for toxicity. The duration of antibiotic treatment ranges from 14 to 21 days based on the severity of the presentation. Once the PR interval has become less than 300 msec, telemetry should be continued for another 24 to 48 h [68].

The first-line treatment recommended by National Institute for Health and Care Excellence (NICE) guidelines is similar. The main differences are the suggested treatment duration, i.e., 21 days for both hemodynamically stable and unstable patients, and the antibiotic regimen in young patients (under the age of 9), for whom intravenous ceftriaxone is preferred in mild forms of carditis [69].

Tetracyclines, such as doxycycline, have been generally contraindicated in children under the age of 8 and pregnant women due to the risk of irreversible dental staining during developmental age. However, the American Academy of Pediatrics currently supports the use of doxycycline in children for up to 21 days, as two new studies have demonstrated none of the children who took a short course of doxycycline developed dental staining [70,71]. Some authors argue, however, that those studies are based on small numbers of patients and that dosage or duration of treatment were less than the one recommended by the Centers for Disease Control and Prevention (CDC) guidelines for erythema migrans (which is also the same for Lyme carditis). Furthermore, there is no evidence of improved outcomes with doxycycline rather than beta-lactam in patients without neurologic involvement or coinfection with *Anaplasma phagocytophilum*. Lastly, doxycycline’s associated side effects, such as nausea, vomiting, diarrhea, and photodermatitis, may reduce compliance in children. Therefore, it is reasonable to evaluate the clinical situation on a case-by-case basis, taking into consideration if doxycycline has been previously administered, as the risk of teeth staining increases with cumulative exposure [68,72].

In pregnant women, antibiotic treatment must be appropriate for the stage of pregnancy. Tetracyclines are usually avoided, especially during the second half of the pregnancy, due to the risk of hepatotoxicity in the mother and permanent discoloration of bones and teeth in the fetus. In these cases, second-choice antibiotics such as macrolides are usually preferred [73], even if caution should be exercised due to the risk of QT interval prolongation [69].

Short-term use of doxycycline during breastfeeding is considered acceptable, but long (>21 days) or repeated courses should be avoided as a theoretical precaution [74]. The concentration of tetracyclines has been reported to be very low in breastfed infants, likely due to drug chelation with calcium in breast milk leading to decreased absorption by the infant [75].

### 7.3. Pacing

Lyme carditis requires both etiologic and supportive therapies to manage possible electrical and hemodynamic complications. Patients with severe forms of Lyme carditis and those presenting with myopericarditis should be hospitalized for continuous heart monitoring [14].

Telemetry is essential, as 35% of patients experience a progression to a high-degree block requiring ventricular stimulation [17]. Fortunately, in most cases, AVB subsides within the first week of antibiotic treatment, so temporary pacing is the preferred modality and permanent pacemaker placement is not indicated for long-term therapy [14].

If 1:1 atrioventricular conduction is not re-established after two weeks of antibiotic therapy, permanent pacemaker implantation should be considered.

Alternatively, if 1:1 atrioventricular conduction is restored within two weeks, an exercise test should be performed, at least after 10 days from hospitalization, to define the point of Wenckebach and assess nodal conduction at higher rates. During incremental stimulation, the point of Wenckebach corresponds to the heart rate, expressed in milliseconds, in which an AVB occurs. If the point of Wenckebach appears at frequencies below 90 rpm, a permanent pacemaker is recommended; whereas if the 1:1 atrioventricular conduction is maintained at a heart rate of more than 120 rpm, the patient may be discharged with the sole oral antibiotic treatment, and repeated ECG should be performed 4–6 weeks after the testing. If 1:1 atrioventricular conduction is lost between 90 and 120 rpm, a re-evaluation after 4–6 weeks may be warranted [24].

## 8. Prognosis and Probability of Reinfection

Cardiac involvement in Lyme disease is usually transient and self-limited, and the prognosis is promising for patients who receive early antibiotic treatment, with most cases resolving within two weeks [76].

More than 90% of high-degree AVB recedes within one week after the beginning of antibiotic treatment [44], and it typically resolves progressing from third-degree AVB, to Luciani–Wenckebach, to first-degree AVB, and finally to the normalization of the atrioventricular conduction [24]. Minor conduction abnormalities usually resolve within six weeks [9].

Rarely, the persistence of atrioventricular conduction disturbances has been described [77,78,79].

Few data are available about long-term outcomes in patients with complete restoration of normal atrioventricular conduction after proper antibiotic treatment. In a single-center observational study, Wang et al. reported a case series of seven patients who completely recovered from Lyme carditis. After a mean follow-up time of 21 months, all patients were asymptomatic, with no evidence of atrioventricular conduction abnormalities (median PR time 168 msec) [80].

Reinfection usually occurs in treated patients with a previous episode of erythema migrans and typically presents as another erythema migrans lesion, while it is less common after a disseminated infection. Particularly, reinfection is rare after late manifestations, such as arthritis, due to the high protection conferred by IgG antibodies directed against decorin-binding protein and other antigens [81].

## 9. Conclusions

Lyme carditis is a rare manifestation of Lyme disease, occurring within weeks to 1–2 months after *Borrelia burgdorferi* infection, and usually presents as conduction abnormalities. Clinical suspicion of Lyme carditis should arise in presence of an AVB in patients with a history of tick exposure in endemic areas, especially in young people. The SILC score is a useful tool to assess the probability of Lyme carditis. If the score reveals an intermediate or high probability of infection, serologic testing is recommended, and antibiotic treatment should be initiated. Patients who are symptomatic or carry an elevated risk of developing a high-degree AVB should be hospitalized and treated with intravenous antibiotics. Telemetry monitoring is crucial, due to the risk of a rapid progression to complete AVB, particularly in patients with a PR interval greater than 300 msec. Temporary pacing may be required in cases of hemodynamic instability. However, in most cases, an adequate antibiotic treatment allows the restoration of normal atrioventricular conduction, so permanent pacing is usually not required. Once restored, to confirm the stability of the atrioventricular 1:1 conduction, it is recommended to perform a pre-discharge exercise stress test.

A prompt diagnosis and treatment are essential to prevent further complications and accelerate the healing process and grant Lyme carditis an excellent prognosis.

## Figures and Tables

**Figure 1 pathogens-11-00582-f001:**
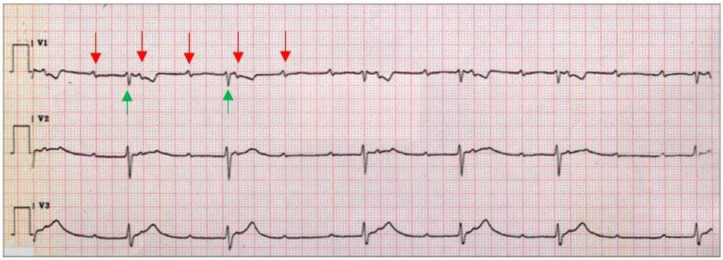
Third-degree AVB in Lyme carditis. The ECG shows the dissociation of atria (red arrows) and ventricles (green arrows), with variable PR intervals.

**Figure 2 pathogens-11-00582-f002:**
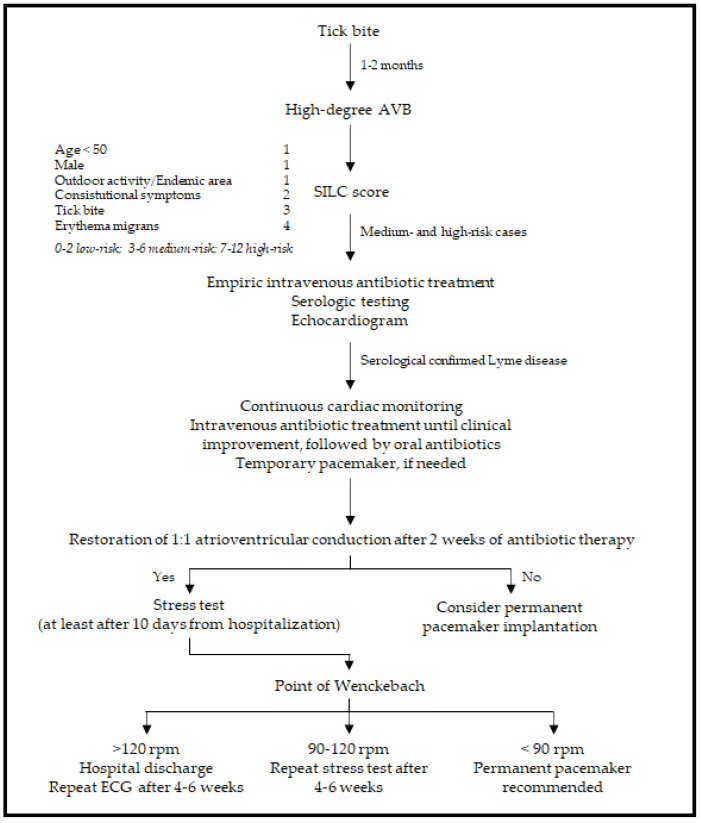
Summary diagram of diagnosis and management of Lyme carditis. Adapted from [24]. Constitutional symptoms are represented by fever, malaise, dyspnea, and arthralgia. SILC = Suspicious Index in Lyme Carditis [44].

**Table 1 pathogens-11-00582-t001:** Intrinsic and extrinsic causes of AVB. Image adapted from 2021 European Society of Cardiology Guidelines on cardiac pacing and cardiac resynchronization therapy [37]. TAVI = transaortic valve replacement.

INTRINSIC	EXTRINSIC
**Infectious Diseases**	**Metabolic Disorders**
Endocarditis (perivalvular abscess) Myocarditis Lyme disease Diphtheria Toxoplasmosis Chagas disease	Hypothyroidism Anorexia Hypoxia Acidosis Hypothermia
**Infiltrative Diseases**	**Electrolyte Imbalance**
Sarcoidosis Amyloidosis Hemochromatosis	Hypokalemia Hyperkaliemia Hypercalcemia Hypermagnesemia
**Collagen Vascular Diseases**	**Neurological Disorders**
Rheumatoid arthritis Scleroderma Systemic lupus erythematosus Storage diseases Neuromuscular diseases	Increased intracranial pressure Central nervous system tumors Temporal epilepsy Obstructive sleep apnea
**Cardiac Surgery**	**Miscellaneous**
Coronary artery bypass grafting Valve surgery (including TAVI) Heart transplant Radiation therapy Intended or iatrogenic AVB	Physical training (sports) Vagal refle XDrug effects Idiopathic paroxysmal AVB
**Others**	
Idiopathic (aging, degenerative) Infarction/ischemia Cardiomyopathies Genetic disorders Congenital heart diseases	

**Table 2 pathogens-11-00582-t002:** Scheme of the recommended antibiotic treatment, according to Centers for Disease Control and Prevention guidelines [67].

**MILD** **(1st-Degree AVB with PR Interval <300 msec)**	ADULTS	**Antibiotic Drug**	**Dosage and Way of Administration**	**Duration (Days)**
		Doxycycline	100 mg, twice per day orally	14–21 days
		Amoxicillin	500 mg, three times per day orally	14–21 days
		Cefuroxime	500 mg, twice per day orally	14–21 days
	CHILDREN	Doxycycline **	4.4 mg/kg per day orally, divided into 2 doses Maximum 100 mg per dose	14–21 days
		Amoxicillin	50 mg/kg per day orally, divided into 3 doses Maximum 500 mg per dose	14–21 days
		Cefuroxime	30 mg/kg per day orally, divided into 2 doses Maximum 500 mg per dose	14–21 days
**SEVERE** **(Symptomatic, 1st-degree AVB with PR interval ≥300 msec, 2nd or 3rd degree AV block)**	ADULTS	Ceftriaxone	2 g intravenously * once a day	14–21 days
	CHILDREN	Ceftriaxone	2 g intravenously * once a day	14–21 days

* After resolution of symptoms and high-grade AV block, consider transitioning to oral antibiotics to complete the treatment course. ** According to NICE guidelines [69], the first-line recommended antibiotic treatment for children under 9 years old is Ceftriaxone intravenously 2 g once a day, for children under 50 kg: 80 mg/kg once per day.

## Data Availability

Not applicable.

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
