# Peer review of "Lyme Carditis: From Pathophysiology to Clinical Management"

_pathogens, 2022, doi:10.3390/pathogens11050582_

Round 1

Reviewer 1 Report

This review provides a comprehensive overview of Lyme carditis, caused by Borrelia burgdorferi sensu lato. The details on the microbiology, transmission cycle and clinical manifestations are all accurate. The authors have included all the standard tests for picking up congenital heart block, other AV blocks, arrhythmias and structural conditions. I found no errors in the treatment regimens. The details on the serology and diagnostic testing are accurate. The limitations are clearly described. 

For the benefit of readers with different expertise, the authors may wish to consider defining what is atrioventricular block and what is meant by first/second/third degree AVB.  Perhaps more detail could be added to the Figure 1 or the caption to explain what is shown to the non-specialist. 

Some minor rephrasing is required on Section 4, paragraph 1 from line 94 to 100. 
The opening sentence "Spirochetes must survive..." feels quite general and may perhaps suggest that other spirochetes are involved (e.g. leptospira, treponema etc). If this is changed to "The spirochetes must survive.." this will be sufficient and consistent with earlier text.

Line 96: "extreme temperatures" suggests an extremophile. Please change this to "a wide range of temperatures" or something with similar meaning. 

On line 100, the authors state that "...an organism completely dependent upon the surrounding environment". To prevent any confusion, the authors may wish to clarify this by adding a sentence to make it clear that the surrounding environment is the host tissue, as Borrelia are obligate parasites and not free living. 

A citation is required on line 428 to support the figure of 94.3% of AVB resolves.

Lines 91 and 441 should be lowercase b in burgdorferi 

Reviewer 2 Report

I had the pleasure to review this excellent review. It is well organized and figures and tables are self explanatory.

Some minor suggestions:

General comments

  1. The only "algorithm" (accepted by CDC) for the diagnosis and management is form reference #21. Please consider asking permission for reproduction or modify it in order to provide the reader with a comprehensive algorithm for management.
  2. The figure adapting the SILC Score should mention the original article by Besant et al (ref #41) in the caption of the figure.
  3. Specific comments:
  4. A frequent question is: What happens to patients after recovering from Lyme carditis? The answer is in this paper, please cite and discuss: Long-term Outcomes in Treated Lyme Carditis. Wang CN, et al. Curr Probl Cardiol. 2021 Jul 24:100939. doi: 10.1016/j.cpcardiol.2021.100939
